# The chromatin remodeler Chd1 supports MRX and Exo1 functions in resection of DNA double-strand breaks

**Marco Gnugnoli** [ID], **Erika Casari** [ID], **Maria Pia Longhese** [ID] *

Dipartimento di Biotecnologie e Bioscienze, Università degli Studi di Milano-Bicocca, Milano, Italy

* mariapia.longhese@unimib.it

**Data Availability Statement:** All relevant data are within the manuscript and its Supporting Information files.

## Abstract

Repair of DNA double-strand breaks (DSBs) by homologous recombination (HR) requires that the 5'-terminated DNA strands are resected to generate single-stranded DNA over-hangs. This process is initiated by a short-range resection catalyzed by the MRX (Mre11-Rad50-Xrs2) complex, which is followed by a long-range step involving the nucleases Exo1 and Dna2. Here we show that the *Saccharomyces cerevisiae* ATP-dependent chromatin-remodeling protein Chd1 participates in both short- and long-range resection by promoting MRX and Exo1 association with the DSB ends. Furthermore, Chd1 reduces histone occupancy near the DSB ends and promotes DSB repair by HR. All these functions require Chd1 ATPase activity, supporting a role for Chd1 in the opening of chromatin at the DSB site to facilitate MRX and Exo1 processing activities.

## Author summary

DNA double strand breaks (DSBs) are among the most severe types of damage occurring in the genome because their faulty repair can result in chromosome instability, commonly associated with carcinogenesis. Efficient and accurate repair of DSBs relies on several proteins required to process them. However, eukaryotic genomes are compacted into chromatin, which restricts the access to DNA of the enzymes devoted to repair DNA DSBs. To overcome this natural barrier, eukaryotes have evolved chromatin remodeling enzymes that use energy derived from ATP hydrolysis to modulate chromatin structure. Here, we examine the role in DSB repair of the ATP-dependent chromatin remodeler Chd1, which is frequently mutated in prostate cancer. We find that Chd1 is important to repair DNA DSBs by homologous recombination (HR) because it promotes the association with a damaged site of the MRX complex and Exo1, which are necessary to initiate HR. This Chd1 function requires its ATPase activity, suggesting that Chd1 increases the accessibility to chromatin to initiate repair of DNA lesions.

## Introduction

DNA double-strand breaks (DSBs) are the most common cause of genomic instability, because their inaccurate repair can lead to chromosomal rearrangements. DSBs can be repaired by

**Funding:** This work was supported by Fondazione AIRC under IG 2017 - ID. 19783 and Progetti di Ricerca di Interesse Nazionale (PRIN) 2017 to M.P. L. E.C. was supported by the Italian Ministry of University and Research (MIUR) through grant "Dipartimenti di Eccellenza-2017" to University of Milano Bicocca. The funders had no role in study design, data collection and analysis, decision to publish, or preparation of the manuscript.

**Competing interests:** The authors have declared that no competing interests exist.

either homologous recombination (HR), which uses homologous DNA from sister chromatids or homologous chromosomes as a template for repair, or non-homologous end joining (NHEJ), which directly re-ligates the broken DSB ends [1].

HR is initiated by nucleolytic degradation of the 5'-terminated strands at the DSB ends to generate 3'-ended single stranded DNA (ssDNA), in a process called resection [2]. In both yeast and mammals, DSB resection is initiated by the Mre11-Rad50-Xrs2/NBS1 (MRX/N) complex that, aided by the Sae2 protein (CtIP in mammals), cleaves the 5'-terminated DNA strand on either side of a DSB [3]. This step is followed by 3'-5' nucleolytic degradation by Mre11, which proceeds back towards the DNA end and by the Exo1 or the Dna2 nuclease, which degrades double-stranded DNA (dsDNA) in the 5'-3' direction [4–11]. The resulting 3'-ended ssDNA is first coated by the Replication Protein A (RPA) complex, which is replaced by the Rad51 recombinase, creating a nucleoprotein filament that searches and anneals to a homologous DNA sequence [1]. Repair can then take place via synthesis-dependent strand annealing (SDSA) or the canonical recombination pathway that involves formation of a double Holliday junction [12].

The repair of DNA DSBs is challenged by the packaging of genomic DNA through histone and non-histone proteins into a high-order structure called chromatin, raising the question as to how the DNA repair machineries overcome this barrier to gain access to damaged DNA. The presence of nucleosomes inhibits DSB resection *in vitro* [13]. Furthermore, a genome-wide analysis of resection endpoints around Spo11-induced DSBs during meiosis showed that resection frequently terminates at nucleosomes, reflecting a tendency for nucleosomes to block nuclease activity *in vivo* [14]. Chromatin immunoprecipitation experiments support nucleosome disassembly near DSBs [15]. Furthermore, recent data indicate that histones exclusively associate with dsDNA and that the rate of histone loss correlates with resection [16], suggesting that nucleosome eviction occurs concomitantly with DSB resection.

Indeed, chromatin structure is tuned by various processes such as nucleosome remodeling by ATP-dependent chromatin remodelers. These protein complexes use the energy derived from ATP hydrolysis to alter histone-DNA interactions, resulting in nucleosome sliding, eviction, and/or histone exchange [17,18]. Several ATP-dependent nucleosome remodelers have been implicated in HR, particularly with regard to DSB resection [19–20]. In budding yeast, the RSC, INO80 and SWI/SNF protein complexes are recruited to chromatin regions adjacent to a nuclease-induced DSB [21–24]. Furthermore, their lack reduces not only nucleosome removal/sliding but also DSB resection [21,22,25–28], suggesting that nucleosome eviction and resection are intrinsically coupled. These changes in chromatin compaction have been shown to facilitate the access to DSBs of DNA repair proteins, such as MRX, Rad51 and Rad52 [15,21,22,25].

Another chromatin remodeler implicated in DSB resection is Fun30 (SMARCAD1 in mammals), which has highest sequence homology to INO80-like remodelers but lacks the split ATPase domain [29–31]. In contrast to INO80 that promotes DSB resection either by removing histones or by controlling distribution of the histone variant H2A.Z adjacent to a DSB [15,21,26,32,33], Fun30 promotes DSB resection by antagonizing the association with DSBs of Rad9 that inhibits the processing activity of Exo1 [29–31].

The evolutionary conserved chromodomain-helicase-DNA-binding protein 1 (Chd1) is an ATP-dependent chromatin remodeler that contains a N-terminal tandemly arranged chromodomain and a central ATPase-helicase domain that confers nucleosome spacing, removal or exchange activity [34,35]. In contrast to most chromatin remodelers, Chd1 is active as a monomer and does not assemble as a multi-subunit complex. Chd1 has the ability to assemble histones along dsDNA and to induce a regular nucleosome spacing [36–39]. In yeast, Chd1 was shown to associate with RNA polymerase II elongation factors on actively transcribed genes

and to be important for recycling histones over coding regions during transcription [40–42]. Experiments in yeasts have shown that Chd1 is also important for generating spaced nucleosomes at the 5' end of several genes [38,43–45].

Only one CHD protein is present in yeast (Chd1), whereas at least nine CHD proteins are expressed in vertebrates. Among them, CHD1, CHD2, CHD3, CHD4, CHD6 and CHD7 have been implicated in the cellular response to DNA damage. In particular, CHD2, CHD3, CHD4 and CHD7 accumulate at DNA regions flanking a DSB and promote the recruitment of proteins involved in NHEJ [46], whereas CHD6 is a key component of the signaling and transcriptional response to reactive oxygen species [47].

In humans, CHD1 is one of the most frequently inactivated genes in prostate cancer [48–50]. Furthermore, its loss sensitizes prostate cancer cells to chemotherapeutic DNA-damaging agents, suggesting a role in the DNA damage response [51]. Consistent with this hypothesis, CHD1 is recruited to UV-damaged nucleosomes in a manner dependent on the DNA binding protein XPC [52]. Furthermore, it promotes the repair of UV-damaged DNA by stimulating the handover between XPC protein and the TFIIH complex at DNA damaged sites [52]. CHD1 is also recruited to chromatin in response to DSBs, where it promotes the loading of CtIP [53,54]. Furthermore, loss of CHD1 decreases the assembly of RPA and RAD51 foci at DNA breaks and stalled replication forks [53,55], suggesting a role in DSB resection. Finally, in *Saccharomyces cerevisiae*, Chd1 interacts with Exo1 and participates in the generation of meiotic crossovers by enabling the processing of joint molecules by both Exo1 and the mismatch repair complex Mlh1-Mlh3 (MutLγ) [56].

In this study, we found that *Saccharomyces cerevisiae* Chd1 improves the efficiency of nucleosome eviction from the DSB ends. Furthermore, it promotes DSB resection by enhancing the association of the MRX complex and Exo1 with the DSB ends. The lack of its ATPase activity impairs all these functions, suggesting that Chd1 promotes MRX and Exo1 resection activities by increasing their accessibility to DSBs.

## Results

### Chd1 is recruited to a DSB and its lack reduces histone removal

To investigate whether Chd1 has a direct role in the repair of DNA DSBs, we first evaluated whether Chd1 is physically enriched at a DSB by chromatin immunoprecipitation (ChIP). To this end, we used a strain background carrying a galactose-inducible HO endonuclease, which generates a single DSB at the *MAT* locus in the presence of galactose [57]. To minimize the effect of DSB repair, the *MAT* homology regions *HML* and *HMR* were deleted, leading to a DSB that cannot be repaired by HR [57]. Following HO induction by galactose addition, Chd1-Myc was recruited near the HO-induced DSB and its binding increases over three hours (Fig 1A).

After a DSB is formed, nucleosomes are rapidly evicted at both sides of the DSB and this process is thought to promote DSB repair by facilitating the access of DNA repair proteins [19,20]. As Chd1 has a nucleosome eviction activity [34], we analyzed occupancy of histone H3 near the HO-induced DSB at both the *LEU2* and the *MAT* loci. To exclude possible effects of DNA replication on histone association with DNA, HO expression was induced by galactose addition to G2-arrested cells that were kept arrested in G2 with nocodazole for the duration of the experiment. Furthermore, to exclude that possible differences in histone occupancy were due to different repair kinetics, repair of the HO-induced DSB at the *LEU2* locus was prevented by deleting *RAD52*, whereas repair of the HO-induced DSB at the *MAT* locus was prevented by deleting the homologous donor loci *HML* and *HMR*. The H3 signal detected near the HO-induced DSB at both the *LEU2* and the *MAT* loci remained higher in *chd1Δ* than in wild type cells, suggesting that Chd1 participates in histone removal near a DSB (Fig 1B).

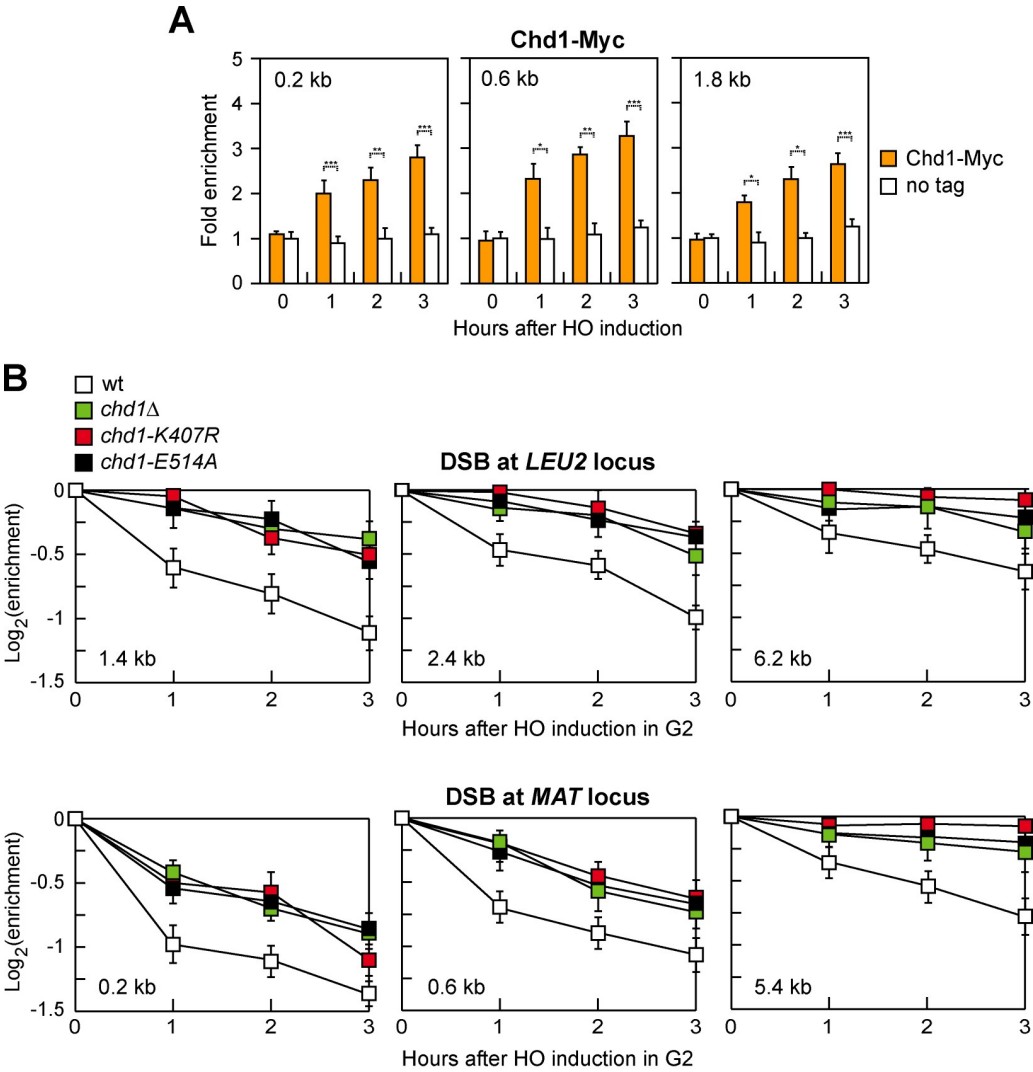

**Fig 1. Chd1 recruitment to a DSB and histone removal.** (A) Exponentially growing YEPR cell cultures of JKM139 derivative strains were transferred to YEPRG, followed by Chd1-Myc ChIP at the indicated distances from the HO-cut site compared to untagged Chd1 (no tag). Data are expressed as fold enrichment at the HO-cut site over that at a non-cleavable locus (*ARO1*), after normalization to the corresponding input for each time point. Fold enrichment was then normalized to cut efficiency. Plotted values are the mean values ± s.d. from three independent experiments. ***$P<0.005$, **$P<0.01$, *$P<0.05$, *t*-test. (B) HO expression was induced by galactose addition to G2-arrested cells carrying the HO system at the *LEU2* or at the *MAT* locus. Cells were kept arrested in G2 by nocodazole throughout the experiment. Histone H3 ChIP with anti-H3 antibody at the indicated distances from the HO-cut site. Data are expressed as fold enrichment at the HO-cut site over that at the non-cleavable *ARO1* locus after normalization to the corresponding input for each time point. Fold enrichment was normalized to cut efficiency. Plotted values are the mean values ± s.d. from three independent experiments.

Chd1 carries an ATP-binding domain (AAA domain) that contains conserved Walker A and B motifs [58]. Within the P-loop of Walker A, a conserved lysine residue (K407 in Chd1) in the consensus sequence GXXXXGK[T/S] (where X is any amino acid) directly interacts with the phosphates of ATP. Mutation of this residue eliminates both ATP binding and ATPase activity [59–61]. The Walker B motif contains aspartate (D513 in Chd1) and glutamate (E514 in Chd1) residues within the hhhhDE sequence (where h represents a hydrophobic amino acid) that are crucial for ATPase activity, with the D residue coordinating Mg$^{2+}$ and the

E residue activating water for the hydrolysis reaction [62]. Mutation of the E residue was shown to impair nucleotide hydrolysis without affecting ATP binding [60,62,63]. To test whether the ATPase activity of Chd1 is required for histone eviction around the DSB, we introduced either the K407R or the E514A amino acid substitution into Chd1. Both *chd1-K407R* and *chd1-E514A* mutant cells were as defective in histone removal from the HO-induced DSB as *chd1Δ* cells (Fig 1B).

## Chd1 promotes DSB resection

Nucleosome eviction from DSBs occurs concomitantly with DSB resection [16], prompting us to monitor directly the generation of ssDNA at the HO-induced DSB in *chd1Δ*, *chd1-K407R* and *chd1-E514A* mutant cells. Because ssDNA cannot be cleaved by most restriction enzymes, generation of ssDNA was assessed by testing resistance to cleavage as resection proceeds beyond restriction sites located at different distances from the HO-cut site at the *MAT* locus (Fig 2A). First, we used a Southern blot analysis approach to detect the appearance of slower migrating bands (r1-r6) after denaturing gel electrophoresis of SspI-digested genomic DNA and hybridization with a probe that anneals to the unresected strand at one side of the DSB (Fig 2B). When HO was induced by galactose addition to exponentially growing cells, the resection products (r2 to r6) appeared less efficiently in galactose-induced *chd1Δ*, *chd1-K407R* and *chd1-E514A* mutant cells compared to wild type cells (Fig 2B and 2C). The resection defect of *chd1Δ* cells was similar to that of *sae2Δ* cells, whereas it was less severe than that of *mre11Δ* cells (S1 Fig).

Detection of SspI-resistant ssDNA by denaturing gel electrophoresis does not allow one to monitor the resection events that do not proceed beyond the SspI site located 0.9 kb from the HO-induced DSB. Furthermore, the signal for the r1 resection product, which can be detected when resection does not proceed beyond the SspI site located 1.7 kb from the DSB, is very low and difficult to quantify. Thus, we used a quantitative PCR-based method to evaluate generation of restriction enzyme-resistant ssDNA [64]. *CHD1* deletion caused a reduction in ssDNA generation very close to the HO-cut site (0.15 kb, 0.65 kb and 0.9 kb) (Fig 2D), indicating a defect in initiation of resection. The same analysis at more distant sites (1.7 kb and 3.5 kb) (Fig 2D) confirmed the long-range resection defect that was detected by denaturing Southern blotting. We can conclude that Chd1 is involved in both short- and long-range resection.

The 3'-ended ssDNA generated during DSB resection is coated by the RPA complex, which is replaced by Rad51 to generate a nucleoprotein filament that invades and anneals to a homologous DNA sequence [1]. Although protein extracts from wild type, *chd1Δ*, *chd1-K407R* and *chd1-E514A* cells contained similar Rad51 amount (Fig 2E), Rad51 association at different distances from the HO-induced DSB was reduced in *chd1Δ*, *chd1-K407R* and *chd1-E514A* cells compared to wild type cells (Fig 2F), consistently with a role of Chd1 in both initiation and extension of DSB resection.

## Chd1 promotes MRX and Exo1 association with DSBs

DSB resection involves sequential action of short- and long-range nucleases. In short-range resection, Mre11 endonuclease, aided by Sae2, cleaves the 5'-terminated DNA strand at ~250–300 nucleotides from the DSB ends, followed by degradation toward the DNA ends by Mre11 exonuclease. Then, Exo1 or Dna2 resects thousands of nucleotides in length in the 5'-3' direction away from the DSB ends [3–11]. While MRX and Sae2 binding to DSBs occurs independently of each other [65], MRX has a structural role in promoting Exo1 and Dna2 association with DSBs [66], thus explaining the more severe resection defect caused by the lack of any MRX subunit compared to that caused by the lack of Mre11 nuclease activity.

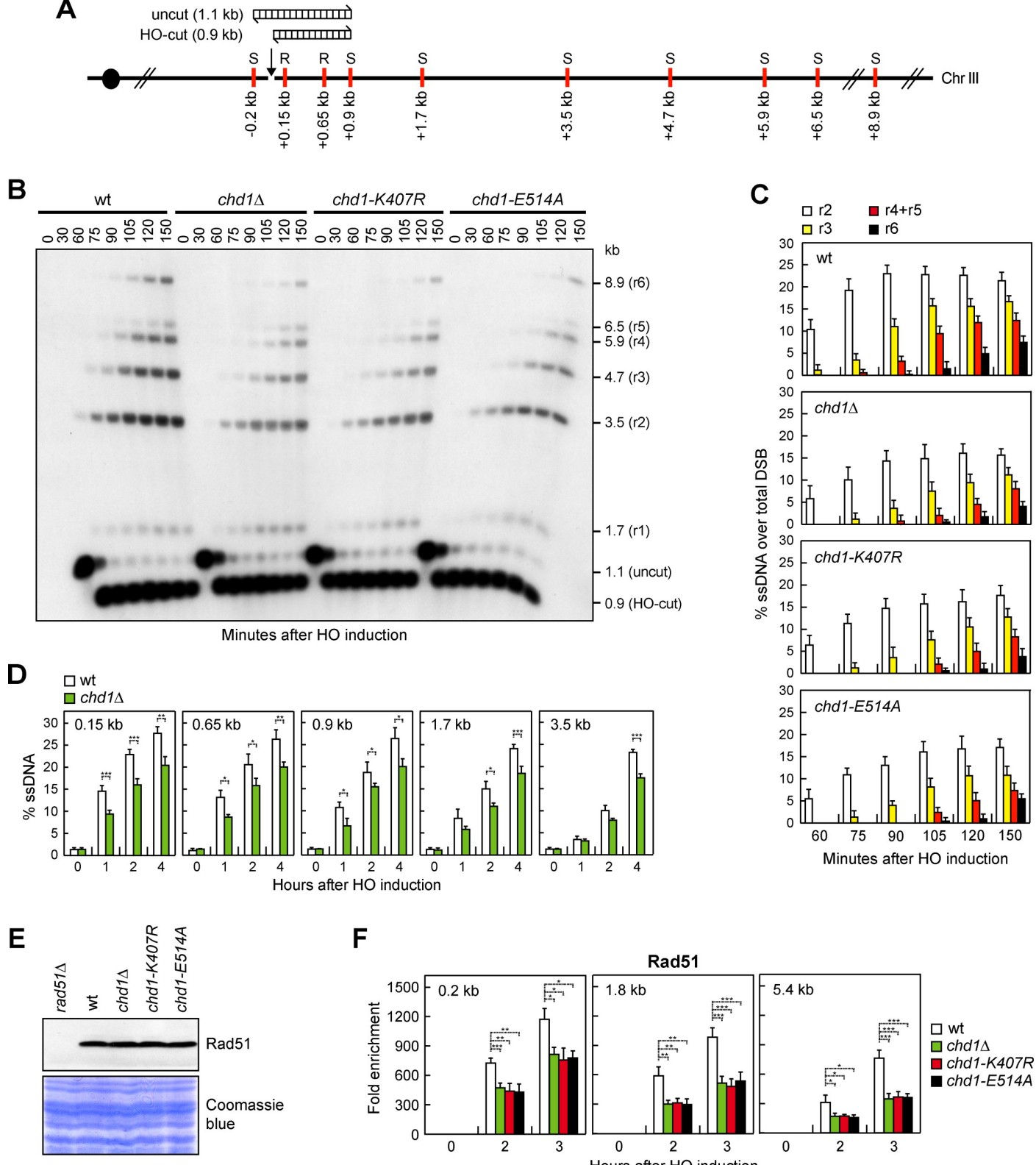

**Fig 2. Chd1 dysfunction reduces DSB resection.** (A) Schematic representation of the *MAT* locus and the distance of RsaI (R) and SspI (S) restriction sites from the HO-cut site. The DNA fragments detected in panel B before (uncut) and after HO cleavage (HO-cut) were also indicated. (B) YEPR exponentially growing cell cultures of JKM139 derivative strains were transferred to YEPRG at time zero. Southern blot analysis of SspI-digested genomic DNA after alkaline gel electrophoresis

with a probe that anneals to the unresected strand. 5'-3' resection progressively eliminates SspI sites (S), producing SspI fragments (r1 through r6) detected by the probe. (C) The experiment as in panel B has been independently repeated three times, and the mean values are represented with error bars denoting s.d. (D) Quantification of ssDNA by qPCR at the indicated distances from the HO-cut site. Plotted values are the mean values of three independent experiments, with error bars denoting s.d. $^{***}P < 0.005$, $^{**}P < 0.01$, $^{*}P < 0.05$, $t$-test. (E) Western blot with anti-Rad51 antibodies of extracts used for the ChIP analysis shown in panel F. The same amount of extracts was separated by SDS-PAGE and stained with Coomassie Blue as loading control. (F) Rad51 ChIP at the indicated distances from the HO-induced DSB. Data are expressed as fold enrichment at the HO-cut site over that at the non-cleavable *ARO1* locus, after normalization to the corresponding input for each time point. Fold enrichment was normalized to cut efficiency. Plotted values are the mean values ± s.d. from three independent experiments. $^{***}P<0.005$, $^{**}P<0.01$, $^{*}P<0.05$, $t$-test.

As Chd1 dysfunction leads to a defect in both short- and long-range resection, we measured MRX, Sae2 and Exo1 association with the HO-induced DSB. The amount of Mre11 (Fig 3A) and Exo1 (Fig 3B) bound at the HO-induced DSB was lower in *chd1Δ*, *chd1-K407R* and *chd1-E514A* cells than in wild type cells. The decreased Mre11 and Exo1 recruitment was not due to lower protein levels, as protein extracts prepared from wild type, *chd1Δ*, *chd1-K407R* and *chd1-E514A* cells contained similar amounts of Mre11 (Fig 3C) and Exo1 (Fig 3D). By contrast, Sae2 association with the HO-induced DSB was similar in wild type, *chd1Δ*, *chd1-K407R* and *chd1-E514A* cells (Fig 3E). Thus, we can conclude that Chd1 facilitates MRX and Exo1 association with DSBs.

## Chd1 promotes DSB repair by HR

The finding that Chd1 promotes DSB resection led us to investigate whether Chd1 has a role in HR. Among the HR repair pathways, single-strand annealing (SSA) is used to repair a DSB flanked by direct DNA repeats when resection uncovers the complementary DNA sequences that can then anneal to each other [67]. To measure the efficiency of SSA, we used YMV45 derivative strains that carry the *GAL-HO* construct and tandem repeats of the *LEU2* gene located 4.6 kb apart on chromosome III, with the HO cutting site adjacent to one of the repeats (Fig 4A) [68]. HO was induced by galactose addition to exponentially growing cells and galactose was maintained in the medium in order to re-cleave the HO sites that can be rejoined by NHEJ. When DSB repair was monitored by Southern blot analysis with a *LEU2* probe, accumulation of the SSA repair product was delayed in *chd1Δ*, *chd1-K407R* and *chd1-E514A* cells compared to wild type cells (Fig 4B and 4C), indicating a role for Chd1 in the SSA repair mechanism. Consistent with a defective DSB repair by SSA, *chd1Δ*, *chd1-K407R* and *chd1-E514A* cells showed a decreased viability on galactose-containing plates (HO expression on) compared to wild type cells (Fig 4D).

Because the SSA repair mechanism does not require strand invasion and therefore does not involve the Rad51 protein [69], we investigated the role of Chd1 in the generation of Rad51-dependent crossover (CO) and non-crossover (NCO) events by ectopic recombination. In the canonical HR pathway, ssDNA invades the homologous dsDNA to form a D-loop structure consisting of heteroduplex DNA and displaced ssDNA. If the displaced ssDNA anneals with the complementary sequence on the other side of the break, extension by DNA synthesis and ligation result in the formation of a double Holliday junction, whose cleavage results in equal number of NCO and CO products. However, if the invading strand extended by DNA synthesis is displaced and anneals with the complementary sequences on the other side of the DSB, this event leads to NCO products by SDSA [70,71].

To analyze formation of CO and NCO products, we used tGI354 derivative strains that carry two copies of the *MAT*a sequence [72]. One copy carries the HO cutting site and is located ectopically on chromosome V, whereas the endogenous *MAT* sequence on chromosome III carries a single base pair mutation that prevents cleavage by HO (*MAT*a-*inc*) (Fig 5A). The HO-induced DSB at the *MAT* sequence on chromosome V can be repaired by using

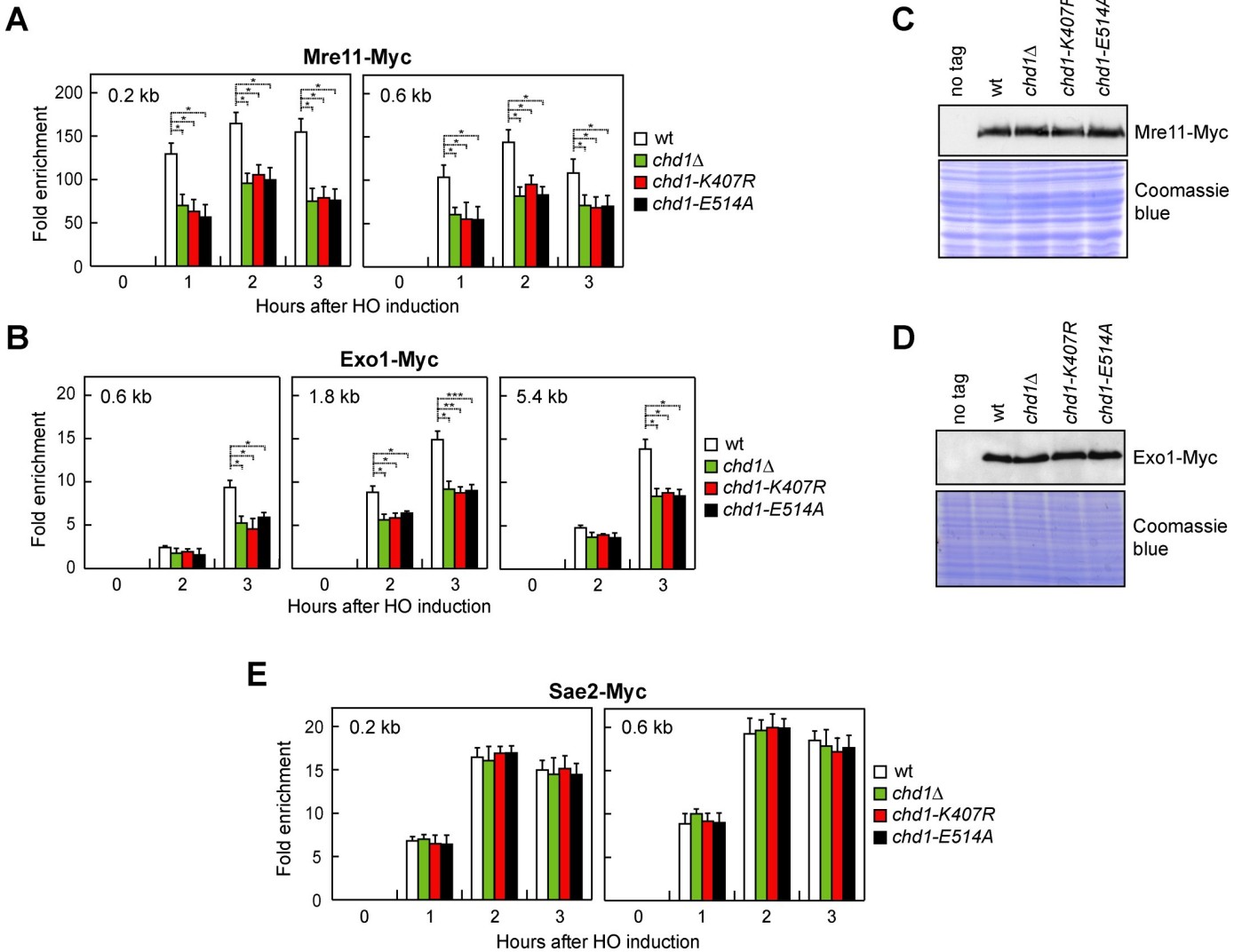

**Fig 3. Chd1 dysfunction impairs MRX and Exo1 association with a DSB.** (A,B) Mre11-Myc (A) and Exo1-Myc (B) ChIP at the indicated distances from the HO-cut site. Data are expressed as fold enrichment at the HO-cut site over that at a non-cleavable locus (*ARO1*), after normalization to the corresponding input for each time point. Fold enrichment was normalized to cut efficiency. Plotted values are the mean values ± s.d. from three independent experiments. ***$P<0.005$, **$P<0.01$, *$P<0.05$, *t*-test. (C,D) Western blot with anti-Myc antibodies of extracts used for the ChIP analysis shown in panels A and B. (E) Sae2-Myc ChIP at the indicated distances from the HO-cut site.

the uncleaved *MAT*a-*inc* sequence on chromosome III, resulting in CO and NCO products (Fig 5A) [72,73]. HO was induced by galactose addition to G2-arrested cells and galactose was maintained in the medium to cleave the HO sites that were eventually reconstituted by NHEJ. Both the 3 kb and the 3.4 kb band resulting from NCO and CO recombination events, respectively, accumulated less efficiently in *chd1Δ* and *chd1-E514A* cells compared to wild type cells, with the NCO band decreasing more severely than the CO band (Fig 5B and 5C), indicating a role for Chd1 in Rad51-dependent HR.

Consistent with defective DSB repair by ectopic recombination, a lower percentage of *chd1Δ* and *chd1-E514A* cells were able to form colonies on galactose-containing plates compared to wild type cells (Fig 5D).

The role of Chd1 in supporting DSB repair appears to be restricted to HR-based mechanisms. In fact, when we measured the ability of cells to re-ligate by NHEJ a plasmid that was

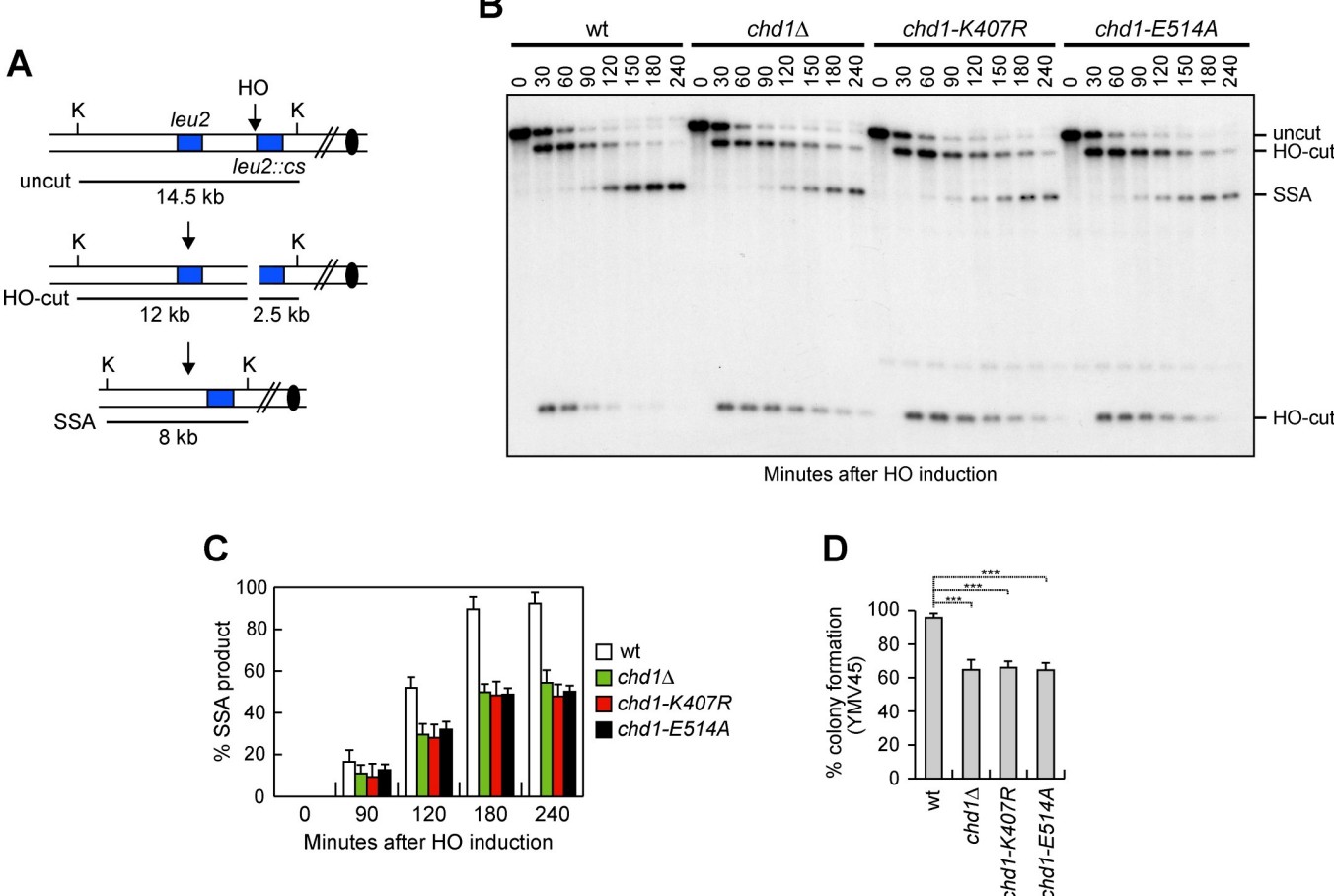

**Fig 4. Chd1 dysfunction reduces DSB repair by SSA.** (A) Schematic representation of the YMV45 chromosome III region, where a unique HO-cut site is adjacent to the *leu2*::cs sequence, which is 4.6 kb apart from the homologous *leu2* sequence. HO-induced DSB results in generation of 12 kb and 2.5 kb DNA fragments (HO-cut) that can be detected by Southern blot analysis with a *LEU2* probe of KpnI-digested genomic DNA. DSB repair by SSA generates a product of 8 kb (SSA). K, KpnI. (B) Exponentially growing YEPR cell cultures of YMV45 derivative strains were transferred to YEPRG. Southern blot analysis of KpnI-digested genomic DNA. (C) Densitometric analysis of the SSA product. Plotted values are the mean values of three independent experiments as in panel B, with error bars denoting s.d. (D) Percentage of colony formation on YEPRG plates relatives to colony formation on YEPD plates. The reported values are the mean values of three independent experiments, with error bars denoting s.d. ***$P<0.005$, *t*-test.

linearized before being transformed into the cells, the efficiency of plasmid re-ligation was similar in wild type, *chd1Δ* and *chd1-E514A* cells (Fig 5E). Furthermore, the amount of Ku70 bound at the HO-induced DSB in both *chd1Δ* and *chd1-E514A* cells was similar to that of wild type cells (Fig 5F).

## Chd1 supports DNA damage resistance and long-range resection when MRX is not fully functional

The *rad50-V1269M* (*rad50-VM*) mutation leads to a decreased MR$^{VM}$X association with DSBs [74]. To investigate whether the diminished MRX binding to DSBs in *chd1Δ* cells is physiologically important, we analyzed the effect of Chd1 dysfunction in *rad50-VM* cells. *CHD1* deletion and the presence of *chd1-K407R* or *chd1-E514A* allele, which caused by themselves a mild sensitivity to high doses of phleomycin (phleo), camptothecin (CPT) or methyl methanesulfonate (MMS) (S2 Fig), exacerbated the sensitivity to genotoxic agents of *rad50-VM* cells (Fig 6A).

As previously reported [74], *rad50-VM* cells slightly decreased Mre11 association with the HO-induced DSB (Fig 6B). Although similar amount of Mre11 can be detected in protein

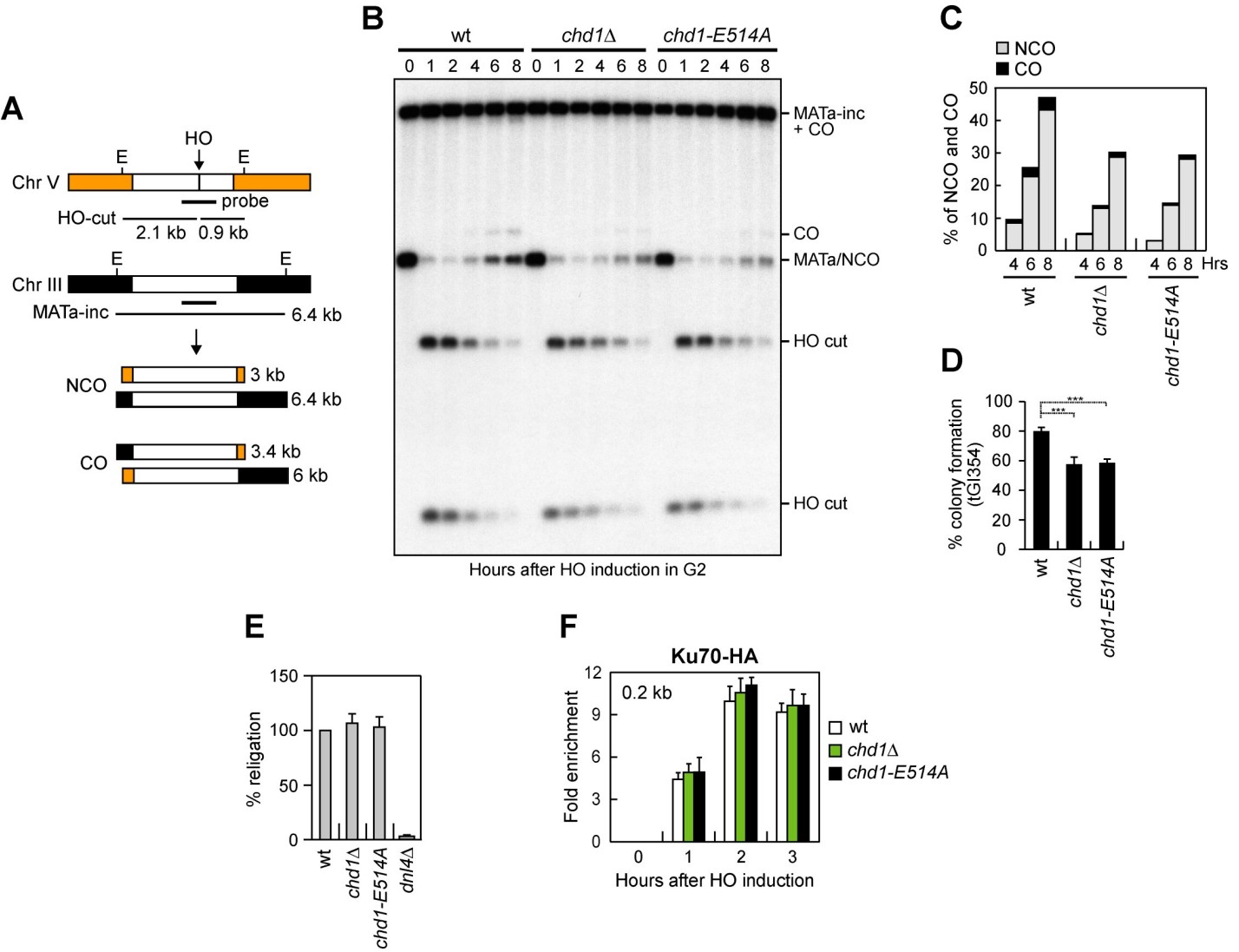

**Fig 5. Chd1 dysfunction reduces DSB repair by HR.** (A) System to detect ectopic recombination. HO generates a DSB at a *MAT*a DNA sequence inserted on chromosome V, while the homologous *MAT*a-*inc* region on chromosome III cannot be cut by HO and is used as a donor to generate noncrossover (NCO) and crossover (CO) products. E, EcoRI. (B) G2-arrested YEPR cell cultures of tGI354 derivative strains were transferred to YEPRG at time zero and were kept arrested in G2 by nocodazole. Southern blot analysis of EcoRI-digested genomic DNA with the *MAT*a probe depicted in panel A. (C) Densitometric analysis of CO (3.4 kb) versus NCO (3 kb) repair bands at the indicated times after HO induction. (D) Percentage of colony formation on YEPRG plates relatives to colony formation on YEPD plates. Plotted values are the mean values of three independent experiments, with error bars denoting s.d. ***$P<0.005$, *t*-test. (E) Plasmid re-ligation assay. Cells were transformed with the same amounts of BamHI-linearized pRS316 plasmid DNA. Data are expressed as percentage of re-ligation relative to wild type that was set up at 100% after normalization to the corresponding transformation efficiency. (F) Ku70-HA ChIP at the indicated distance from the HO-cut site.

extracts from wild type, *chd1Δ*, *rad50-VM* and *chd1Δ rad50-VM* cells (Fig 6C), the amount of Mre11 bound to the HO-induced DSB was lower in *chd1Δ rad50-VM* cells than in *chd1Δ* and *rad50-VM* cells (Fig 6B), thus explaining the increased DNA damage sensitivity of *chd1Δ rad50-VM* double mutant cells.

We also monitored the resection kinetics by following resistance to cleavage by restriction enzymes at different distances from the HO-induced DSB. As previously reported [74], the *rad50-VM* mutation affected DSB resection only very mildly (Fig 6D). ssDNA generation close to the HO-cut site (0.15 kb and 0.65 kb) in *chd1Δ rad50-VM* cells was similar to that of *chd1Δ* cells (Fig 6D). By contrast, *chd1Δ rad50-VM* cells showed a reduction in ssDNA generation at

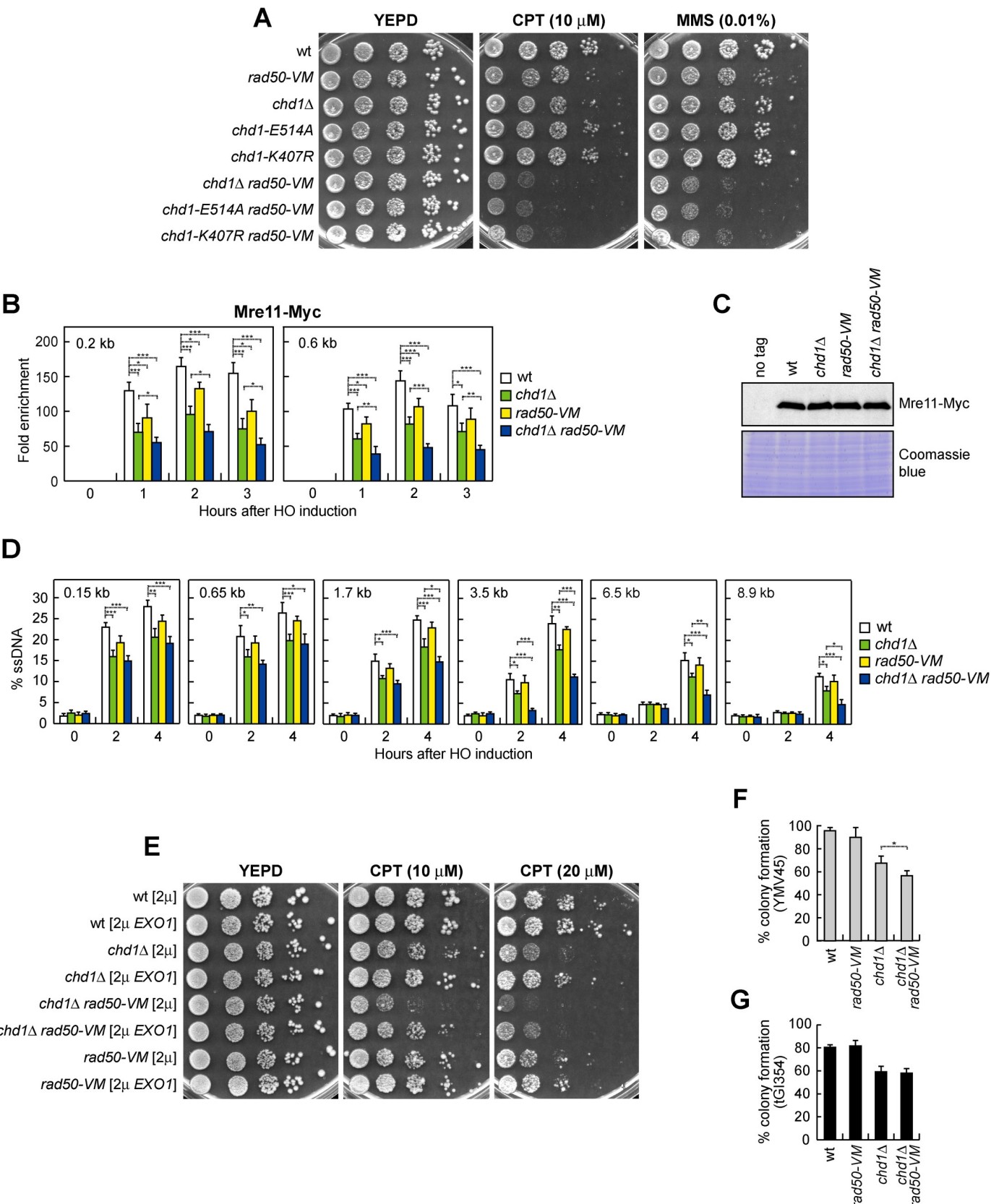

**Fig 6. Chd1 dysfunction exacerbates the DNA damage sensitivity and the long-range resection defect of *rad50-VM* cells.** (A) Exponentially growing cultures were serially diluted (1:10) and each dilution was spotted out onto YEPD plates with or without CPT or MMS. (B) Mre11-Myc ChIP at the indicated distances from the HO cleavage site. Data are expressed as fold enrichment at the HO-cut site over that at a non-cleavable locus (*ARO1*), after normalization to the corresponding input for each time point. Fold enrichment was normalized to cut efficiency. Plotted values are the mean values ± s.d. from three independent experiments. ***$P<0.005$, **$P<0.01$, *$P<0.05$, *t*-test. (C) Western blot with anti-Myc antibodies of extracts used for the ChIP analysis shown in panel B. (D) Quantification of ssDNA by qPCR at different distances from the HO-cut site. Plotted values are the mean values of three independent experiments, with error bars denoting s.d. ***$P<0.005$, **$P<0.01$, *$P<0.05$, *t*-test. (E) Exponentially growing cultures were serially diluted (1:10) and each dilution was spotted out onto YEPD plates with or without CPT. (F,G) Percentage of colony formation of YMV45 (F) and tGI354 (G) derivative strains on YEPRG plates relative to colony formation on YEPD plates. Plotted values are the mean values of three independent experiments, with error bars denoting s.d. *$P<0.05$, *t*-test.

more distant sites (1.7 kb, 3.5 kb, 6.5 kb and 8.9 kb) compared to both *chd1Δ* and *rad50-VM* cells (Fig 6D), indicating a more severe long-range resection defect in the double mutant. Consistent with a role of Chd1 in promoting extended DSB processing, the DNA damage sensitivity of both *chd1Δ* and *chd1Δ rad50-VM* cells was partially suppressed by *EXO1* overexpression (Fig 6E), indicating that part of their DNA damage sensitivity was due to defects in DNA processing.

While DSB repair by SSA requires that resection reaches the complementary DNA sequences that can anneal to each other, DSB repair by ectopic recombination does not require extensive processing of the DSB ends [75–77]. Consistent with the finding that *chd1Δ rad50-VM* cells compromised extended DSB resection more severely than *chd1Δ* cells, percentage of survival of *chd1Δ rad50-VM* cells was lower than that of *chd1Δ* cells upon generation of a HO-induced DSB that is repaired by SSA (Fig 6F). By contrast, *chd1Δ* and *chd1Δ rad50-VM* cells showed similar percentage of survival upon generation of a HO-induced DSB that is repaired by ectopic recombination (Fig 6G).

## Discussion

In mammals, loss of the ATP-dependent chromatin-remodeling protein CHD1 impairs DSB repair and decreases the assembly of RPA and RAD51 foci [51], suggesting a role for this protein in DSB resection. This study shows that the lack of *S. cerevisiae* Chd1 reduces nucleosome eviction from the DSB ends. Furthermore, its lack impairs both short- and long-range resection by reducing MRX and Exo1 association with DSBs. The Chd1 functions in nucleosome eviction from DSBs and resection require its ATPase activity, suggesting that Chd1 promotes resection by acting as nucleosome evictor in the opening of chromatin to promote/stabilize MRX and Exo1 association with DSBs. Consistent with a role of chromatin compaction in counteracting MRX and Exo1 accessibility to the DSB ends, assembly of DNA into nucleosomes causes inhibition of both Exo1 resection activity and MRX-dependent activation of Tel1 kinase [13,78]. Furthermore, high-throughput single-molecule microscopy has shown that MRN searches for free DNA ends by one-dimensional facilitated diffusion and transiently dissociates from the DNA backbone to bypass a nucleosome [79].

The decreased DSB resection in *chd1* mutants impairs DSB repair by SSA, which requires that resection of the DSB ends reaches the complementary DNA sequences. Furthermore, the lack of Chd1 or of its ATPase activity leads to a reduction of NCO products during ectopic recombination. Consistent with a previous finding that the lack of Chd1 does not affect formation of ectopic CO products in mitotically dividing cells [56], the generation of CO products in *chd1* mutants is only slightly affected. In any case, the role of Chd1 appears specific for DSB repair by HR, as the lack of Chd1 affects neither DSB repair by NHEJ nor the association of the Ku complex with DSBs.

The function of Chd1 in promoting MRX binding/persistence to DSBs becomes important to support DNA damage resistance when MRX accumulation at DSBs is suboptimal, such as in the presence of the *rad50-VM* mutation. This mutation reduces MRX association with DSBs

and the lack of Chd1 reduces further the amount of MR$^{VM}$X bound to DSBs. As a consequence, *chd1Δ rad50-VM* cells are more sensitive to genotoxic agents compared to each single mutant. Interestingly, while resection very close to the DSB occurs with similar kinetics in *chd1Δ* and *chd1Δ rad50-VM* cells, long-range resection is compromised more severely in *chd1Δ rad50-VM* cells compared to both *chd1Δ* and *rad50-VM* single mutants. These findings, together with the observation that overexpression of *EXO1*, which is the main nuclease involved in long-range resection, partially suppresses the DNA damage sensitivity of both *chd1Δ* and *chd1Δ rad50-VM* cells, suggests a direct role of Chd1 in supporting Exo1 resection activity. In accord with this hypothesis, Chd1 was found to interact with Exo1 and to enable MutLγ-Exo1-dependent processing of joint molecules into COs during meiosis I [56].

While in mammals CHD1 was shown to promote the recruitment of CtIP to DSBs, the association of Sae2, the yeast CtIP counterpart, does not require Chd1 function. However, it should be pointed out that the localization of CtIP to DSBs in both mammals and *S. pombe* requires the MRN complex [80–82], whereas Sae2 association with DSBs in *S. cerevisiae* occurs independently of MRX [65]. As the role of mammalian CHD1 in promoting MRN association with DSBs has not been investigated yet, one possibility is that the poor CtIP binding to DSBs in CHD1-depleted cells might be due to a diminished MRN association with DSBs.

In conclusion, we propose that Chd1 increases the accessibility of chromatin to facilitate/ stabilize the association of MRX and Exo1 with DSBs, which in turn initiate DSB processing. The *CHD1* gene is frequently mutated in prostate cancer where these mutations are associated with a poor prognosis [48–51]. Our finding that Mre11 dysfunction can be rendered synthetically lethal with *chd1* mutations in the presence of genotoxic agents suggests that MRX inhibitors in combination with DNA-damaging chemotherapy could be beneficial in patients whose tumors are defective in CHD1 function.

## Materials and methods

### Yeast strains and media

Strain genotypes are listed in S1 Table. Strains JKM139, YMV45 and tGI354, used to detect DSB resection, DSB repair by SSA and DSB repair by ectopic recombination, respectively, were kindly provided by J. Haber (Brandeis University, Waltham, USA). Cells were grown in YEP medium (1% yeast extract, 2% bactopeptone) supplemented with 2% glucose (YEPD), 2% raffinose (YEPR) or 2% raffinose and 3% galactose (YEPRG). Gene disruptions were generated by one-step PCR homology cassette amplification and standard yeast transformation method.

### Spot and DSB survival assays

For spot assays, exponentially growing cell cultures were diluted to $1 \times 10^7$ cells/ml. 10-fold serial dilutions were spotted on YEPD with or without the indicated DNA damaging drugs. Plates were incubated for 3 days at 30˚C. To determine viability in DSB assays, cells exponentially growing in YEPR were plated onto YEPD and YEPRG plates. Survivor colonies were counted after 3 days of incubation at 30˚C, and the survivor percentage was calculated by normalizing colony number on YPRG to colony number on YEPD.

### DSB resection at the *MAT* locus

DSB end resection at the *MAT* locus in JKM139 derivative strains was analyzed on alkaline agarose gels by using a single-stranded probe that anneals to the unresected DSB strand, as previously described [83]. Quantification of DSB resection was determined by calculating the ratio of band intensities for ssDNA to the total amount of DSB products. To normalize to cut

efficiency, the value of the uncut band was subtracted from the total amount of DSB products for each time point. Quantitative PCR (qPCR) analysis of DSB resection at the *MAT* locus in JKM139 derivative strains was carried out as previously described [64]. Genomic DNA was extracted at different time points following HO induction. Oligonucleotides were designed to detect ssDNA at specific distances from the DSB (0.15 kb, 0.65 kb, 0.9 kb, 1.7 kb, 3.5 kb, 6.5 kb and 8.9 kb) (S2 Table). The DNA was digested with both SspI and RsaI restriction enzymes. A mock reaction without the restriction enzymes was set up in parallel. qPCR was performed on both digested and mock samples using SsoFast EvaGreen supermix (Bio-Rad) on the Bio-Rad CFX Connect Real-Time System apparatus. For each time point, Ct values were normalized to those obtained from the mock sample, and then further normalized to values obtained from an amplicon in the *KCC4* control gene. Finally, the obtained values were normalized to the HO-cut efficiency measured by qPCR by using oligonucleotides that anneal on opposite sides with respect to the HO cutting sequence (S2 Table). The percentage of HO-cut was calculated by comparing the Ct values before and after HO induction in undigested samples.

## DSB repair by SSA

DSB repair by SSA in YMV45 strains was detected by Southern blot analysis using an Asp718-SalI fragment containing part of the *LEU2* gene as a probe, as previously described [84]. Quantitative analysis of DSB repair by SSA was determined by calculating the ratio of band intensities for SSA to the total amount of SSA and DSB products for each time point. To normalize to cut efficiency, the value of the uncut band was subtracted from the total amount of SSA and DSB products.

## DSB repair by ectopic recombination

DSB repair by ectopic recombination was detected in tGI354 background as previously described [84]. To determine the repair efficiency, the intensity of the uncut band at 2 h after HO induction (maximum efficiency of DSB formation) was subtracted from the normalized values of NCO and CO bands at the subsequent time points after galactose addition. The obtained values were divided by the normalized intensity of the uncut *MAT*a band at time zero before HO induction (100%).

## Chromatin immunoprecipitation and qPCR

ChIP analysis was performed with anti-HA (12CA5), anti-Myc (9E10), anti-H3 (ab1791, Abcam) and anti-Rad51 (ab63798, Abcam) antibodies as previously described [74]. Quantification of immunoprecipitated DNA was achieved by qPCR on a Bio-Rad CFX Connect Real-Time System apparatus. Triplicate samples in 20 µl reaction mixture containing 10 ng of template DNA, 300 nM for each primer, 2X SsoFast EvaGreen supermix (1725201, Bio-Rad) (2X reaction buffer with dNTPs, Sso7d-fusion polymerase, MgCl$_2$, EvaGreen dye, and stabilizers) were run in white 96-well PCR plates Multiplate (MLL9651, Bio-Rad). The qPCR program was as follows: step 1, 98˚C for 2 min; step 2, 90˚C for 5 s; step 3, 60˚C for 15 s; step 4, return to step 2 and repeat 45 times. At the end of the cycling program, a melting program (from 65˚C to 95˚C with a 0.5˚C increment every 5 s) was run to test the specificity of each qPCR. For each time point, data are expressed as fold enrichment at the HO-cut site over that at the non-cleaved *ARO1* locus, after normalization of each ChIP signals to the corresponding input signals. Fold enrichment was normalized to cut efficiency that was determined by qPCR. For histone loss, log2 values of the relative enrichment were calculated. Oligonucleotides used for qPCR analyses are listed in S2 Table.

## Plasmid religation

The centromeric pRS316 plasmid was digested with the BamHI restriction enzyme before being transformed into the cells. Parallel transformation with undigested pRS316 DNA was used to determine the transformation efficiency. Efficiency of re-ligation was determined by counting the number of colonies grown on medium selective for the plasmid marker and normalizing them with respect to the transformation efficiency for each sample. The re-ligation efficiency in mutant cells was compared to that of wild type cells that was set up to 100%.

## Western blotting

Protein extracts for western blot analysis were prepared by trichloroacetic acid (TCA) precipitation. Frozen cell pellets were resuspended in 200 μL 20% TCA. After the addition of acid-washed glass beads, the samples were vortexed for 10 min. The beads were washed with 200 μL of 5% TCA twice, and the extract was collected in a new tube. The crude extract was precipitated by centrifugation at 3000 rpm for 10 min. TCA was discarded and samples were resuspended in 70 μL 6X Laemmli buffer (60mM Tris pH 6.8, 2% SDS, 10% glycerol, 100mM DTT, 0.2% bromophenol blue) containing 0.9% 2-mercaptoethanol and 30 μL 1M Tris pH8.0. Prior to loading, samples were boiled at 95˚C and centrifuged at 3.000 rpm for 10 min. To detect Mre11-Myc, Exo1-Myc and Rad51, TCA protein extracts were separated on 10% polyacrylamide gels and probed with anti-Myc (9E10) or anti-Rad51 (ab63798, Abcam) antibody.

## Supporting information

**S1 Table. List of yeast strains used in this study.**
(DOCX)

**S2 Table. List of oligonucleotides used in this study.**
(DOCX)

**S1 Fig. DSB resection.** YEPR exponentially growing cell cultures of JKM139 derivative strains were transferred to YEPRG at time zero. Southern blot analysis of SspI-digested genomic DNA after alkaline gel electrophoresis with a probe that anneals to the unresected strand. 5'-3' resection progressively eliminates SspI sites (S), producing SspI fragments (r1 through r6) detected by the probe.
(TIF)

**S2 Fig. DNA damage sensitivity of *chd1* mutants.** Exponentially growing cultures were serially diluted (1:10) and each dilution was spotted out onto YEPD plates with or without CPT, MMS or phleomycin.
(TIF)

**S1 Data. Original data sheets.**
(XLSX)

## Acknowledgments

We thank J. Haber for yeast strains and S. Ratti for preliminary data.

## Author Contributions

**Conceptualization:** Marco Gnugnoli, Maria Pia Longhese.

**Data curation:** Marco Gnugnoli.

**Formal analysis:** Marco Gnugnoli, Erika Casari.

**Funding acquisition:** Maria Pia Longhese.

**Investigation:** Marco Gnugnoli, Erika Casari.

**Methodology:** Marco Gnugnoli.

**Project administration:** Maria Pia Longhese.

**Supervision:** Maria Pia Longhese.

**Validation:** Marco Gnugnoli.

**Visualization:** Marco Gnugnoli, Erika Casari, Maria Pia Longhese.

**Writing – original draft:** Maria Pia Longhese.

**Writing – review & editing:** Marco Gnugnoli, Erika Casari, Maria Pia Longhese.

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
