## [Decision Letter · Decision Letter 0]

24 May 2021

Dear Dr Longhese,

Thank you very much for submitting your Research Article entitled 'The chromatin remodeler Chd1 supports the MRX complex in resection of DNA double-strand breaks' to PLOS Genetics. The manuscript was fully evaluated at the editorial level and by three independent peer reviewers. The reviewers appreciated the attention to an important problem, but raised concerns about the current manuscript that need to be addressed. For example, survival data following DSB induction needs to be included and there was an issue raised with regard to data normalization. In addition, the current work needs to be put in the context of what is know about the roles of other chromatin remodelers during DSB repair. Based on the reviews, we will not be able to accept this version of the manuscript, but we would be willing to review a much-revised version. We cannot, of course, promise publication at that time.

If you decide to revise the manuscript for further consideration at PLOS Genetics, please aim to resubmit within the next 60 days, unless it will take extra time to address the concerns of the reviewers, in which case we would appreciate an expected resubmission date by email to plosgenetics@plos.org.

[LINK]

We are sorry that we cannot be more positive about your manuscript at this stage. Please do not hesitate to contact us if you have any concerns or questions.

Yours sincerely,

Sue Jinks-Robertson, Ph.D.

Associate Editor

PLOS Genetics

Gregory P. Copenhaver

Editor-in-Chief

PLOS Genetics

Reviewer's Responses to Questions

**Comments to the Authors:**

Reviewer #1: review attached

Reviewer #2: The authors document that Chd1 has a role in resection of DSB ends and in promoting efficient repair of DSBs. Chd1 is shown to promote MRX binding to DSB ends. Chd1 likely works in the same pathway with MRX as deletion of CHD1 increases the sensitivity of rad50VM mutant (decreased association with DSB) but not with complete deletion of rad50. Finally, Chd1 is shown to help in nucleosome eviction at DSB ends. Together the manuscript presents interesting set of observations, however the authors need to provide additional evidence to support their conclusions.

Comments

Is the effect Chd1 deletion on resection direct or indirect? As chd1 is a chromatin remodeler it is possible that it regulates protein levels of MRX or other resection enzymes. Is Chd1 recruited to DSBs as would be expected if it played a direct role? What is the evidence for direct involvement of Chd1 in resection? Are Mre11, Rad50, and Xrs2 levels and localization similar in wild type and chd1? Genetics indicates that Chd1 works in the same pathway with MRX so at least one could test few things related to MRX levels and localization.

It looks that mostly initial resection is impacted in chd1 mutants. Authors should comment on this and test whether Chd1 affects also extensive resection.

Page 7

Please specify that chd1 cells are only "mildly" sensitive to "high" doses DNA damage inducing agents. In lower doses there is no sensitivity (Fig 6c).

Page 8-9 “Chd1 promotes DSB repair”

Viability of wt and chd1 cells (tested by plating on YEPGAL plates) in response to DSB needs to be shown for both SSA and ectopic recombination assays to conclude that Chd1 promotes DSB repair. What is shown is only kinetics of repair in nocodazole blocked cells, not the final repair. chd1 mutants are barely sensitive to DNA damaging agents therefore it is surprising that Chd1 is important for the repair of a single DSB. Are chd1 cells sensitive to zeocin, bleomycin or phleomycin?

In case resection was analyzed in nocodazole blocked cells, at least one set of experiments should be done in growing cells.

Comparison of the sensitivity of chd1, rad50 and double mutant in Figure 5D need to be shown across broad range of drugs concentration to support epistasis.

Other comments

Page 11

Chd1 promotes MRX but not Ku recruitment. Authors conclude that it is consistent

with the role of Chd1 in HR and not NHEJ. This should be rephrased as MRX is needed for NHEJ in budding yeast.

Previously it was reported that Chd1 deficient cells have normal resection PMID: 22960743. This was tested in the same strain background and no decrease of initial or extensive resection was observed. What is the reasons of these different observations? Were the cells arrested here in nocodazole? Methods section directs the reader to a manuscript that potentially indicates G2/M nocodazole arrest. If so it would be important to perform one experiment in growing cells (to exclude unlikely possibility that the effect of CHD1 deletion is specific to G2/M arrested cells). In general it should be indicated whether resection was analyzed in nocodazole blocked cells.

Reviewer #3: This is a generally clear and careful study of the role of the chromatin remodeler Chd1 in resection of DSBs in yeast. The overall conclusions seem justified, but there are a few places where the paper would benefit from further clarification.

1. The authors conclude that Chd1 plays some role in promoting the association of MRX with DSB ends. Wouldn’t the same results be obtained if the function of Chd1 was to stabilize its association or prevent its removal? Is there a way to see how long MRX says associated with a DSB?

2. I always admire the clear denaturing gels published by this lab! However, when I look at the results, it seems to me that r1 (1.7 kb) in the mutants is not that different from WT, suggesting to me that longer-range resection is more affected? So maybe a ChIP or Rad51 at a greater distance would be useful? More on resection below.

3. It would be useful in Figs. 1, 2 and 3 to compare these results to mre11∆ or rad50∆ and sae2∆. Is the effect of chd1∆ much less? My impression from Symington or Haber is that deleting MRX delays but doesn’t much reduce HR and even SSA over short distances. For example, Mimitou and Symington’s 2008 SSA results with sae2∆ (Fig. 1) shows a 65% efficiency but little change in kinetics.

4. Why a semi-log plot in Fig. 4?

5. In the absence of Mre11 or Rad50, is Sae2 ChIP affected? How is Sae2 recruited if not through MRX?

6. Again, is chd1∆ rad50-VM less severe than deleting Rad50?

7. Finally, Chd1 has been implicated in late recombinational steps in meiosis by Matos (i.e. in crossovers). There, Chd1 showed meiosis-specific MutSgamma-Exo1 interactions. These data aren’t discussed, but they raise the possibility that Exo1 might be involved in Chd1’s association mitotically as well.

**Have all data underlying the figures and results presented in the manuscript been provided?**

Reviewer #1: Yes

Reviewer #2: Yes

Reviewer #3: Yes

PLOS authors have the option to publish the peer review history of their article (what does this mean?). If published, this will include your full peer review and any attached files.

Reviewer #1: No

Reviewer #2: No

Reviewer #3: No

---

## [Decision Letter · Decision Letter 1]

16 Aug 2021

Dear Dr Longhese,

Thank you very much for submitting your revised Research Article entitled 'The chromatin remodeler Chd1 supports MRX and Exo1 functions in resection of DNA double-strand breaks' to PLOS Genetics. The original three reviewers appreciated the care taken in revising the manuscript and were uniformly positive about the changes. Although there remain a few minor comments that need to be addressed (all editorial in nature), subsequent re-reveiw will not required.  Once you upload a revised version, I'll be happy to accept it.  

1) Briefly summarize your responses to the review comments and a description of the changes you have made in the manuscript.

[LINK]

Yours sincerely,

Sue Jinks-Robertson, Ph.D.

Associate Editor

PLOS Genetics

Gregory P. Copenhaver

Editor-in-Chief

PLOS Genetics

Reviewer's Responses to Questions

**Comments to the Authors:**

Reviewer #1: The article is much improved and the authors took great care to address all the reviewers' comments. THis is greatly appreciated. It was useful and wise to move the data that was hard to interpret to supplemental material and to focus on the data that make sense. The ssDNA at close distances was important to provide.

I have only minor corrections to suggest:

line 74: Fun30 is not really very close to the INO80 family, which is characterized by a split ATPase domain. SWR1 and INO80 both have the split ATPase but Fun30 does not. It is true that Flaus et al. NAR 2006, and Seeber at al 2013 both place Fun30 in the INO80 family based on overall sequence homology, but I think line 74 should be phrased slightly differently: "... Fun30 (SMARCAD1 in mammals) has highest sequence homology to INO80-like remodelers but lacks the split ATPase domain."

line 161 "does not allow one to "

line 184 and elsewhere: " While MRC and Sae2 association to..." Should be either "binding to" or "association with" not as written. This error is made multiple times (see also line 195)

line 193: "contained similar amounts...."

line 312: "Chd1 was found to interact with Exo1 and to enable..." not "to interacts"

The rest is clear and worthy of publication in PLOS Genetics. A very nice paper.

Reviewer #2: The authors performed a number of additional experiments and addressed most of the questions raised by reviewers. Thus my opinion is to accepts revised manuscript.

Reviewer #3: The authors have made a substantial effort to address the concerns of the reviewers. I have only a few minor comments/suggestions:

1. Define the primers and restriction sites used to determine cutting efficiency. I didn’t find them in the text.

2. Whether Chd1 is really involved in histone eviction may depend on your definition. Pfander’s recent paper (Peritore et al. 2021) argues persuasively that there is no histone occupancy on ssDNA. So one might argue that monitoring histone loss is no different from monitoring resection. Other papers do suggest that histones are evicted prior to resection (most recently Tipuraneni et al. 2021, but earlier papers by Sang Lee and Mary Anne Osley). I think to really demonstrate that Chd1 is evicting histones would require that the histone removal be more extensive than the resection and that this difference would go away when Chd1 were absent.

3. One other thing that seems confusing to me: why does SSA not continue to increase beyond 180 min in Fig. 4? If resection is slower but there is more time, should SSA “catch up”? Why is viability lower if there is plenty of time for a viable colony-forming unit to be created? Looking at the HO-cut bands in Fig. 4B, it seems that resection is continuing between 180 and 240 min, but product formation doesn’t increase. This is strange.

4. Fig. 5 shows that deleting Chd1 didn’t affect the crossovers associated with the ectopic recombination strain tGI354. Although the Wild et al. (Matos lab) paper is now cited, the fact that Wild et al. already came to the same conclusion about crossovers in mitotic cells is not acknowledged.

5. At least mention the alternative suggestion that Chd1 might act to stabilize MRX association to DSBs rather than promote its recruitment. (One might tell the difference by using an AID mutant to inactivate Chd1 after recruitment of MRX had occurred; but I am not suggesting this experiment needs to be done).

**Have all data underlying the figures and results presented in the manuscript been provided?**

Reviewer #1: Yes

Reviewer #2: Yes

Reviewer #3: Yes

PLOS authors have the option to publish the peer review history of their article (what does this mean?). If published, this will include your full peer review and any attached files.

Reviewer #1: No

Reviewer #2: No

Reviewer #3: **Yes: **James E. Haber

---

## [Editor Report · Decision Letter 2]

6 Sep 2021

Dear Dr Longhese,

We are pleased to inform you that your manuscript entitled "The chromatin remodeler Chd1 supports MRX and Exo1 functions in resection of DNA double-strand breaks" has been editorially accepted for publication in PLOS Genetics. Congratulations!

Yours sincerely,

Sue Jinks-Robertson, Ph.D.

Associate Editor

PLOS Genetics

Gregory P. Copenhaver

Editor-in-Chief

PLOS Genetics

Comments from the reviewers (if applicable):

**Data Deposition**

http://datadryad.org/submit?journalID=pgenetics&manu=PGENETICS-D-21-00513R2

**Press Queries**

---

## [Editor Report · Acceptance letter]

9 Sep 2021

PGENETICS-D-21-00513R2 

The chromatin remodeler Chd1 supports MRX and Exo1 functions in resection of DNA double-strand breaks 

Dear Dr Longhese, 

We are pleased to inform you that your manuscript entitled "The chromatin remodeler Chd1 supports MRX and Exo1 functions in resection of DNA double-strand breaks" has been formally accepted for publication in PLOS Genetics! Your manuscript is now with our production department and you will be notified of the publication date in due course.

With kind regards,

Katalin Szabo

PLOS Genetics

On behalf of:
